# Heart Rate in Patients with SARS-CoV-2 Infection: Prevalence of High Values at Discharge and Relationship with Disease Severity

**DOI:** 10.3390/jcm10235590

**Published:** 2021-11-28

**Authors:** Alessandro Maloberti, Nicola Ughi, Davide Paolo Bernasconi, Paola Rebora, Iside Cartella, Enzo Grasso, Deborah Lenoci, Francesca Del Gaudio, Michela Algeri, Sara Scarpellini, Enrico Perna, Alessandro Verde, Caterina Santolamazza, Francesco Vicari, Maria Frigerio, Antonia Alberti, Maria Grazia Valsecchi, Claudio Rossetti, Oscar Massimiliano Epis, Cristina Giannattasio

**Affiliations:** 1Cardiology 4, “A.De Gasperis” Cardio Center, ASST GOM Niguarda Cà Granda, 20162 Milan, Italy; michela.algeri@ospedaleniguarda.it (M.A.); sara.scarpellini@ospedaleniguarda.it (S.S.); cristina.giannattasio@unimib.it (C.G.); 2School of Medicine and Surgery, University of Milano-Bicocca, 20126 Milan, Italy; Iside.cartella@ospedaleniguarda.it (I.C.); enzo.grasso@ospedaleniguarda.it (E.G.); 3Rheumatology, Multispecialist Medical Department, ASST GOM Niguarda Ca’ Granda, 20162 Milan, Italy; nicola.ughi@ospedaleniguarda.it (N.U.); Deborah.lenoci@ospedaleniguarda.it (D.L.); francesca.delgaudio@ospedaleniguarda.it (F.D.G.); 4Bicocca Bioinformatics, Biostatistics and Bioimaging Centre—B4, School of Medicine and Surgery, University of Milano Bicocca, 20126 Milan, Italy; davide.bernasconi@unimib.it (D.P.B.); paola.rebora@unimib.it (P.R.); grazia.valsecchi@unimib.it (M.G.V.); oscar.epis@ospedaleniguarda.it (O.M.E.); 5Cardiology 2, “A.De Gasperis” Cardio Center, ASST GOM Niguarda Ca’ Granda, 20162 Milan, Italy; enrico.perna@ospedaleniguarda.it (E.P.); alessandro.verde@ospedaleniguarda.it (A.V.); Maria.frigerio@ospedaleniguarda.it (M.F.); 6Territorial Cardiology “A.De Gasperis” Cardio Center, ASST GOM Niguarda Ca’ Granda, 20162 Milan, Italy; Caterina.Santolamazza@ospedaleniguarda.it (C.S.); francesco.vicari@ospedaleniguarda.it (F.V.); antonia.alberti@ospedaleniguarda.it (A.A.); 7Nuclear Medicine, ASST GOM Niguarda Ca’ Granda, 20162 Milan, Italy; claudio.rossetti@ospedaleniguarda.it

**Keywords:** heart rate, SARS-CoV-2, infection severity, COVID-19

## Abstract

The most common arrhythmia associated with COronaVIrus-related Disease (COVID) infection is sinus tachycardia. It is not known if high Heart Rate (HR) in COVID is simply a marker of higher systemic response to sepsis or if its persistence could be related to a long-term autonomic dysfunction. The aim of our work is to assess the prevalence of elevated HR at discharge in patients hospitalized for COVID-19 and to evaluate the variables associated with it. We enrolled 697 cases of SARS-CoV2 infection admitted in our hospital after February 21 and discharged within 23 July 2020. We collected data on clinical history, vital signs, laboratory tests and pharmacological treatment. Severe disease was defined as the need for Intensive Care Unit (ICU) admission and/or mechanical ventilation. Median age was 59 years (first-third quartile 49, 74), and male was the prevalent gender (60.1%). 84.6% of the subjects showed a SARS-CoV-2 related pneumonia, and 13.2% resulted in a severe disease. Mean HR at admission was 90 ± 18 bpm with a mean decrease of 10 bpm to discharge. Only 5.5% of subjects presented HR > 100 bpm at discharge. Significant predictors of discharge HR at multiple linear model were admission HR (mean increase = β = 0.17 per bpm, 95% CI 0.11; 0.22, *p* < 0.001), haemoglobin (β = −0.64 per g/dL, 95% CI −1.19; −0.09, *p* = 0.023) and severe disease (β = 8.42, 95% CI 5.39; 11.45, *p* < 0.001). High HR at discharge in COVID-19 patients is not such a frequent consequence, but when it occurs it seems strongly related to a severe course of the disease.

## 1. Introduction

The arrhythmic risk related to COronaVIrus-related Disease (COVID) is still under evaluation [1,2]. The most common arrhythmia related to SARS-CoV-2 infection is sinus tachycardia, with palpitations as the principal clinical presentation [3], that sometimes remains after the acute phase of severe illness as a long-term alteration. However, to the best of our knowledge, both the prevalence of high Heart Rate (HR) at hospital discharge and of its persistence over time (and for how long) is currently unknown [4].

HR, and even more its variability, has been related to worse outcomes in infection [5,6]. In fact, it is among the parameters used in some prognostic score for sepsis such as the Sepsis-related Organ Failure Assessment (SOFA) [7]. Despite this, its pharmacological lowering as a therapeutic target in septic patients has not been associated with an improvement in cardiac function [8] nor with the amelioration of mortality risk [9].

From this evidence, one could speculate that HR in infection is simply a marker of a severe clinical condition and a high systemic response to sepsis at presentation. However, another hypothesis could arise, i.e., that it is related to the emerging of an autonomic dysfunction [10]. In fact, the persistence of sinus tachycardia and palpitation in subjects suffering from SARS-CoV-2 infection has been hypothesized to be related to a long-term dysregulation of autonomic system [11].

The importance of HR in diseases has been not only recognized in infection. In fact, an HR higher than 100 bpm has been also inserted into the latest European guidelines on hypertension as a prognostic marker since it is related to future cardiovascular events [12]. Therefore, the primary purpose of the present study was to assess the prevalence of elevated HR at discharge (defined as an HR higher than 100 bpm [13,14]) in patients hospitalized for COVID. Furthermore, we evaluated the variables associated with discharge HR. This study arises from our clinical experience on COVID-19 patients in which we observed a high prevalence of sinus tachycardia and elevated discharge HR that could also persist over time. This led us to design this analysis on patients admitted to our hospital and to try to understand why some patients experience this problem and some others don’t.

## 2. Materials and Methods

### 2.1. Study Population

This monocentric retrospective observational cohort study was performed by reviewing the medical electronical case records of patients who were admitted to Niguarda Hospital after the first Italian autochthonous COVID case on 21 February 2020, until 23 July 2020. Niguarda is one of the largest General Hospital (1167 beds) in the North of Milan within a Metropolitan Area of 3,279,944 inhabitants on 1 January 2020, and it hosts all the medical and surgical disciplines for adults and children including a 24-hours ED with 96.588 visits and 32.612 in hospital admissions covering every intensity of care in 2019.

Inpatients from low-intensity general and specialist medical units, who were consecutively admitted to Niguarda Hospital between 21 February and 8 June 2020, were screened for eligibility if the nasopharyngeal swab to search for the SARS-CoV-2 genome was positive, regardless of the presence of the respiratory disease. The SARS-CoV-2 infection was defined if the genome was detected by reverse transcriptase-polymerase chain reaction (RT-PCR) for one or more out of three SARS-CoV-2 genes tested on at least one nasopharyngeal swab. All the adult patients (≥18 years) with a diagnosis of SARS-CoV-2 infection were included unless the following exclusion criteria were met: current pregnancy, in-hospital death, discharge after the end of the period of observation (23 July 2020), absence of respiratory symptoms related to SARS-CoV-2 infection. If a patient had been re-hospitalised in the period of interest only data from the first hospitalization have been included in the analysis.

### 2.2. Outcome and Predictor Measurements

HR, the main outcome, was measured with central cardiac auscultation for 1 min. It is expressed in beats per minute and the first value at admission and the last before the discharge were collected for analysis purposes. High HR was defined as a discharge HR > 100 bpm [13,14]. The patient’s past medical history, symptoms and signs, vital signs, and laboratory exams at the time of the admission were considered as potential predictors. In detail, the inpatient’s electronical medical records were used to collect data about age, gender, the onset date of COVID symptoms, the presence of COVID pulmonary disease, the need for intensive care, the length of hospitalization, Charlson Comorbidity Index, therapies with lopinavir/ritonavir, azithromycin, hydroxychloroquine or chloroquine, and/or tocilizumab, concomitant drugs acting on heart rate (i.e., beta-blockers, antiarrhythmic, ivabradine, and/or digoxin), systolic and diastolic Blood Pressures (BP), body temperature in degrees Celsius, peripheral oxygen saturation. Moreover, the following laboratory variables were collected: white blood cell count (10^3^/mm^3^), haemoglobin (g/dL), C-reactive protein levels (mg/dL, normal reference < 0.5). Severe disease was defined as the need for Intensive Care Unit (ICU) admission and/or mechanical ventilation.

### 2.3. Compliance with Ethical Standards

The study was conducted as part of the monocentric retrospective non-interventional epidemiological research on Niguarda inpatients with SARS-CoV-2 infection (NOS_COVID-19/1 protocol), which obtained the favourable opinion of the local Ethics Committee (Milano Area 3, register number 249-13052020). The study protocol complies with the Declaration of Helsinki.

### 2.4. Statistical Analysis

Baseline characteristics of patients included in the study were described using numbers and percentages for categorical variables or median and first and thirs quartiles (Q1, Q3) for continuous variables. The distribution of parameters at admission and discharge was reported as medians with Q1, Q3 and with mean and standard deviations (SD) and the mean variation from admission to discharge was estimated with the corresponding 95% confidence interval. The distribution of the HR (bpm) at admission, at discharge, and their variation was compared between patients with or without severe disease (i.e., need for admission to ICU and/or mechanical ventilation) by *t*-test and graphically using boxplots. The distribution of variables at admission or during hospital stay was compared between patients with HR at discharge ≤ 100 bpm vs. >100 bpm using Chi-square test for categorical variables or Mann-Whitney test for continuous variables. The association of variables at admission or during hospital stay with HR at discharge, treated as a continuous variable, was also investigated using linear regression. One model for each predictor was fitted adjusting only for HR at admission. Moreover, a multiple model including as covariates all variables, except treatments for COVID-19, was fitted. A logistic multipredictor regression model with the same covariates and using HR > 100 bpm at discharge as outcome was also fitted. Finally, an enhanced multiple linear model including also treatments for COVID-19 was considered.

## 3. Results

### 3.1. Population Characteristics

Figure 1 showed the flow-chart of enrolled patients. Starting from 957 hospitalizations in our hospital from 21 February 2020 to 8 June 2020. 260 were excluded with a final sample of 697 subjects. Reason for exclusion were: hospitalization before February 21st (*n* = 5), ongoing hospitalization at July 23th (*n* = 13), death (*n* = 205), pregnancy (*n* = 5), age < 18 years (11), hospitalization following the index one (*n* = 9), absence of SARS-CoV-2 related symptoms (*n* = 13) and missing value of the heart rate at discharge (*n* = 4).

Table 1 reports the demographic and clinical characteristics of the enrolled population. Patients showed a median age of 59 (Q1, Q3: 49, 74) years, and a prevalence of males of 60.1%. Symptoms begin 7 days before admission (Q1, Q3: 4, 10) and median hospital stay was 15 (Q1, Q3: 10, 25) days. 84.6% of the subjects showed a SARS-CoV-2 related pneumonia with 13.2% that presented a severe disease (defined as the need for admission to ICU and/or mechanical ventilation) and 13.3% that had a thrombotic event (deep vein thrombosis and/or pulmonary embolism) during the hospitalization.

Median Charlson Comorbidity Index was 2 (Q1, Q3: 1, 4), 46 (6.6%) subjects had a previous history of myocardial infarction, 32 (4.6%) of Heart Failure, 54 (7.7%) of permanent atrial fibrillation and 41 (5.9%) of Chronic Obstructive Pulmonary Disease.

Table 2 reports cardio-circulatory and biochemical variables at admission, at discharge and their variation. Mean HR at admission was 90 ± 18 bpm with a mean variation of 10 bpm while Systolic Blood Pressure and Diastolic one were respectively 133 ± 23 and 75 ± 12 mmHg. Respective mean variation from admission to discharge were 12 and 2 mmHg. Oxygen saturation significantly increased (1.9%, 95% CI: 1.6:2.2; from 95 ± 4 to 97 ± 2%) and body temperature decreased (−0.9 °C, 95% CI: −1.0; −0.8; from 37.0 ± 1.0 to 36.1 ± 0.4 °C).

Mean Haemoglobin, C-reactive protein and white blood cell count at admission were 13.2 ± 1.9 g/dL, 6.7 ± 6.9 mg/dL and 7.66 ± 4.73 10^3^/mm^3^ respectively with a significant decrease during hospitalization.

Regarding therapies (Table 1), most of the patients were treated with hydroxychloroquine (70.0%) while 36.9% were treated with lopinavir/ritonavir, 24.0% with azithromycin, 39.9% with corticosteroid and 10.5% with tocilizumab. 31.0% of the subjects were treated with drugs acting on HR, in particular 28.8% were on beta-blockers, 5.7% on antiarrhythmic and 2.3% on digoxin while only 3 patients took ivabradine.

### 3.2. Elevated Heart Rate at Discharge

When subjects were classified according to discharge HR with the cut-off at 100 bpm (≤100 bpm vs. >100 bpm—Table 1) we found that only a small percentage (5.5%) of subjects presented a high discharge HR and that those subjects were younger (54—Q1, Q337-65 vs. 60—Q1, Q350-74, years, *p* = 0.004) and more frequently presented a severe disease (36.8 vs. 11.8%, *p* < 0.001) with a longer hospital stay (21—Q1, Q310.5-34 vs. 15 Q1, Q39-24, days, *p* = 0.036).

Figure 2 showed the differences in admission (panel A) and discharge (panel B) HR and changes in HR during hospital stay (panel C) when subjects were divided accordingly to the presence of a severe disease. Particularly, subjects with severe disease presented a slightly higher baseline HR (91—Q1, Q380-104 vs. 88—Q1, Q377-100, bpm *p* = 0.024) and a higher discharge HR (87.5 Q1, Q379.5-99 vs. 77—Q1, Q370-86, bpm, *p* < 0.001) with a lower HR decrease during hospitalisation (5—Q1, Q3-10;18 vs. 10—Q1, Q3-1;23, bpm, *p* = 0.005). The prevalence of HR > 100 bpm was significantly higher in subjects with severe disease compared with those without (15.2 vs. 4%, *p* < 0.001).

Furthermore, subjects with high discharge HR presented a lower Charlson Comorbidity Index (a score of 1—Q1, Q30-3 vs. 2 Q1, Q31-4, *p* = 0.002) and higher HR (95—Q1, Q380-104 vs. 88—Q1, Q377-101, bpm, *p* = 0.037) and body temperature (37.2—Q1, Q336.5-38 vs. 36.7—Q1, Q336.1-37.6, °C, *p* = 0.044) at admission as well as higher C-reactive protein (8.4—Q1, Q34.7-13.3 vs. 4.3—Q1, Q31.4-9.3, mg/dL, *p* = 0.002).

No significant differences were seen regarding COVID-19 treatment although it seems that subjects with high discharge HR were treated more frequently with tocilizumab (21.2 vs. 9.9%, *p* = 0.055). Similarly, no differences were seen regarding drugs acting on HR.

### 3.3. Multipredictor Analysis

Table 3 showed the predictors of discharge HR. When adjusted only for baseline HR, a significant relationship was found with age (mean increase = β = −0.10 per year, 95% CI −0.16; −0.04, *p* < 0.001), Charlson Comorbidity Index (β = −0.77 per unit, 95% CI −1.13; −0.41, *p* < 0.001), systolic BP (β = −0.041 per mmHg, 95% CI −0.081; −0.001, *p* = 0.046), hemoglobin (β = −0.52 per g/dL, 95% CI −1.02; −0.03, *p* = 0.039), severe disease (β = 9.64, 95% CI 6.94; 12.35, *p* < 0.001), and HR acting drug (β = −2.67, 95% CI −4.73; −0.62, *p* = 0.011).

When the model was fully adjusted, only admission HR (β = 0.17 per bpm, 95% CI 0.11; 0.22, *p* < 0.001), hemoglobin (β = −0.64 per g/dL, 95% CI −1.19; −0.09, *p* = 0.023), and severe disease (β = 8.42, 95% CI 5.39; 11.45, *p* < 0.001) when significantly associated with discharge HR.

Finally, no significant associations (nor at the model adjusted for baseline HR, nor at the fully adjusted model) were seen with COVID-19 therapies (Appendix A).

When a logistic regression model with the same covariates was performed with the presence of sinus tachycardia as the dependent variable, subjects with severe disease presented an Odds Ratio of 3.783 (95% CI 1.539–9.299, *p* = 0.004) to have a discharge HR > 100 bpm (Appendix A).

## 4. Discussion

Our study showed that high HR at discharge (i.e., sinus tachycardia) is not such a frequent problem. In fact, only 5.5% of the patients discharged from our hospital after COVID-19 present an HR higher than 100 bpm. The second important result of our study is that the discharge HR is strongly related to the evidence of a severe disease that in this analysis was defined as the need for ICU admission and/or mechanical ventilation. In fact, the prevalence of sinus tachycardia at discharge is almost four times higher in patients with severe disease than in patients without severe disease (15.2% vs. 4%). As already mentioned in the introduction, sinus tachycardia is considered the most common arrhythmia related to SARS-CoV-2 infection although no previous definitive data have been published [3]. Some clinicians focused the attention on the persistence of sinus tachycardia over time with symptoms that could remain for longer than 3 weeks (long-COVID) or over 12 weeks also called chronic-COVID [15,16]. However, few data exist, and our paper provides a piece of information on this topic by reporting that only about one over 18 patients was discharged with sinus tachycardia. However, when the analysis was stratified for disease severity, the prevalence increased to about one over 7 patients with severe disease. This last result raised the attention to the most severe patients in which this problem is probably more frequent.

Our study has a major limitation that needs to be immediately discussed, i.e., no HR data following the discharge are available and so we cannot conclude on the question regarding how long this problem persists over time in patients suffering for long-term symptoms. At the best of our knowledge, no paper has been published regarding this point.

The problem of persistent sinus tachycardia after severe respiratory infection is something already evaluated during the Severe Acute Respiratory Syndrome (SARS) coronavirus in 2003. Some paper reports of a prevalence of high HR at 2 months from disease onset between 15% [17] and 38% [18]. In the first paper, the presence of high HR was related to a more severe disease at presentation (similarly to our results) while this was not the case for the second one.

Another point of our study that deserved to be mentioned is the possible mechanism under this increase in HR in some subjects. One could argue that discharge HR is simply a marker of a worst clinical condition at presentation and a higher systemic response to sepsis. However, some papers [11,19,20] proposed another hypothesis, i.e., that long-term symptoms, particularly fatigue and tachycardia, could be related to secondary autonomic dysfunction. In fact, it has been described also to be related to other viral and bacterial infection [21,22] and possible mechanisms of damage are the cytokine storm [23], autoantibody formation [24], a direct viral damage [11] or all the previous together.

Sympathetic hyperactivation is a part of the autonomic dysregulation and it could be one of mechanism at the basis of acute and chronic COVID manifestation [25,26]. In fact, sympathetic overdrive with reduction of vagal anti-inflammatory tone has been proposed as a factor that could help explain the acute COVID-related mortality [26]. Similar condition of imbalance between sympathetic and parasympathic tone has been described also in many CV condition [27,28,29]. These diseases are also the most frequent comorbidities in patients who died of COVID-19 [30] in which, probably, the infection leads to a further enhancement of sympathetic activation. The infection itself and the cytokine storm can further deteriorate the autonomic function leading to an enhancement of the inflammatory response and to negative outcomes [25,26].

HR is a simple and quick way to evaluate sympathetic activity. Although some concern remains on its capacity to discern between different component of the autonomic nervous system, it can be used for a first screening [31].

A more complex but complete way to analyse the autonomic function is HR variability (HRV). Beat-to-beat HRV is a physiological metric that provides insight into the interplay between the parasympathetic and sympathetic nervous systems [32]. A reduction in HRV is a marker of sympathetic overdrive and autonomic dysfunction and, in the set of CV diseases, has been linked to an increase mortality [33]. It has been evaluated also in COVID patients in which two studies found a significant reduction in HRV (and so the presence of sympathetic overdrive) in infected patients [34,35].

The use of drugs acting on HR is quite frequent (31%) with most of these patients taking B-blockers (28.8%). This is probably determined by the prevalence of previous heart disease (6.6% of previous myocardial infarction, 4.6% of congestive heart failure, 7.7% of permanent atrial fibrillation). However, analyses were corrected for drugs acting on HR, and this was confirmed to be not a factor interfere discharge HR in the multivariate model.

Our study has some limitations. The principal one is the absence of a follow-up that could help us defining for how long the sinus tachycardia persists after discharge. The second limitation is the absence of a complete assessment of autonomic function and impairment. In fact, we can only interfere some data from HR hypothesizing an autonomic nervous system involvement. Finally, characterization of immune response or complication was not complete and some other biomarker such as Interleukin, troponin or pro-Brain Natriuretic Peptide, could be useful to complete the analysis regarding the association between HR and the disease severity.

## 5. Conclusions

In conclusion, high HR at discharge (>100 bpm, i.e., sinus tachycardia) in COVID-19 patients is not such a frequent problem and it involves the 5.5% of our population. It seems to be strongly related to the evidence of a severe course of disease. However, post-discharge follow-up data are needed in order to understand persistence over time of sinus tachycardia and if this have a prognostic implication or is only the marker of a worst disease. Finally, more data are also needed to confirm the hypothesis that an autonomic system dysfunction is implicated in a higher HR at discharge.

## 6. Future Research

Post-discharge follow-up with HR evaluation is needed in order to understand persistence over time of sinus tachycardia.Furthermore, from post-discharge HR evaluation and subsequent mortality also information on its prognostic implication could be derived.A better autonomic function assessment should be done in the acute phase and post-discharge in order to confirm the hypothesis that autonomic dysfunction could be implicated in persistent sinus tachycardia.

## Figures and Tables

**Figure 1 jcm-10-05590-f001:**
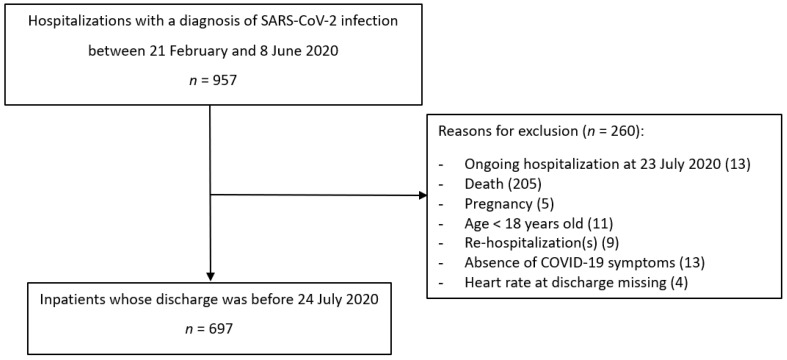
Study flow-chart of patients included in the analysis.

**Figure 2 jcm-10-05590-f002:**
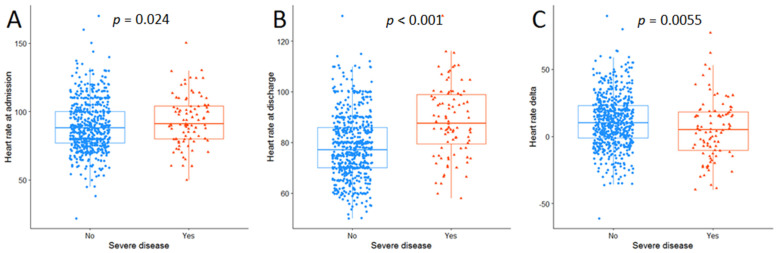
Comparison of the distributions of heart rate (bpm) between patients with or without severe disease at admission (**A**), at discharge (**B**) and their variation (**C**). Vertical lines show 5th and 95th percentiles, boxes show 25th and 75th percentiles, horizontal lines show 50th percentile.

**Table 1 jcm-10-05590-t001:** Features of the hospitalized patients with SARS-CoV-2 infection included in the study and comparisons between patients with heart rate lower than or equal to 100 and higher than 100 beats per minute at hospital discharge.

	Overall *n* = 697	HR ≤ 100 bpm 659 (94.5%)	HR > 100 bpm 38 (5.5%)	*p* *
Gender, female, *n* (%)	278 (39.9)	266 (40.4)	12 (31.6)	0.365
Age, years, median (Q1, Q3)	59 (49, 74)	60 (50, 74)	54 (37, 65)	0.004
Symptom onset before hospital admission, days, median (Q1, Q3)	7 (4, 10)	7 (4, 10)	7 (4, 10)	0.545
COVID-19 pulmonary disease, *n* (%)	590 (84.6)	558 (84.7)	32 (84.2)	0.99
Severe disease, *n* (%)	92 (13.2)	78 (11.8)	14 (36.8)	<0.001
Thrombotic event, *n* (%)	93 (13.3)	85 (12.9)	8 (21.1)	0.233
Length of hospitalization, days, median (Q1, Q3)	15 (10, 25)	15 (9, 24)	21.5 (10.5, 34)	0.036
Charlson Comorbidity Index, median (Q1, Q3)	2 (1, 4)	2 (1, 4)	1 (0, 3)	0.002
History of myocardial infarction, *n* (%)	46 (6.6)	45 (6.8)	1 (2.6)	0.497
Congestive heart failure, *n* (%)	32 (4.6)	32 (4.9)	0 (0.0)	0.32
Permanent atrial fibrillation, *n* (%)	54 (7.7)	53 (8.0)	1 (2.6)	0.368
Chronic obstructive pulmonary disease, *n* (%)	41 (5.9)	39 (5.9)	2 (5.3)	0.99
COVID-19 treatments	517 (74.2)	493 (74.8)	24 (63.2)	0.160
Lopinavir/ritonavir, *n* (%)	257 (36.9)	245 (37.2)	12 (31.6)	0.601
Azithromycin, *n* (%)	167 (24.0)	160 (24.3)	7 (18.4)	0.531
Hydroxychloroquine or chloroquine, *n* (%)	488 (70.0)	466 (70.7)	22 (57.9)	0.135
Tocilizumab, *n* (%)	73 (10.5)	65 (9.9)	8 (21.1)	0.055
Corticosteroids, *n* (%)	278 (39.9)	261 (39.7)	17 (44.7)	0.653
Drugs acting on heart rate	216 (31.0)	209 (31.7)	7 (18.4)	0.123
Beta-blockers, *n* (%)	201 (28.8)	194 (29.4)	7 (18.4)	0.203
Antiarrhythmic, *n* (%)	40 (5.7)	39 (5.9)	1 (2.6)	0.625
Ivabradine, *n* (%)	3 (0.4)	3 (0.5)	0 (0.0)	0.99
Digoxin, *n* (%)	16 (2.3)	15 (2.3)	1 (2.6)	0.99
Heart rate at admission, bpm, median (Q1, Q3)	88 (78, 101)	88 (77, 101)	95 (80, 104)	0.037
Oxygen saturation at admission, %, median (Q1, Q3)	96 (93, 98)	96 (93, 98)	94 (92, 97)	0.253
Systolic blood pressure at admission, mmHg, median (Q1, Q3)	130 (120, 149)	130 (120, 150)	123 (119, 140)	0.174
Diastolic blood pressure at admission, mmHg, median (Q1, Q3)	75 (70, 80)	75 (70, 80)	80 (70, 80)	0.862
Body-temperature at admission, Celsius, median (Q1, Q3)	36.8 (36.1, 37.7)	36.7 (36.1, 37.6)	37.2 (36.5, 38.0)	0.044
Hemoglobin at admission, g/dL, median (Q1, Q3)	13.5 (12.1, 14.5)	13.5 (12.2, 14.6)	12.9 (11.9, 13.8)	0.099
C-reactive protein at admission, mg/dL, median (Q1, Q3)	4.6 (1.4, 9.5)	4.3 (1.4, 9.3)	8.4 (4.7, 13.3)	0.002
White blood cell count at admission, 10^3^/mm^3^, median (Q1, Q3)	6.64 (4.84, 9.25)	6.53 (4.82, 9.31)	7.77 (5.62, 9.20)	0.16
Oxygen saturation at discharge, %, median (Q1, Q3)	97 (96, 98)	97 (96, 98)	97 (95, 98)	0.052
Systolic blood pressure at discharge, mmHg, median (Q1, Q3)	120 (110, 130)	120 (110, 130)	120 (110, 128.75)	0.169
Diastolic blood pressure at discharge, mmHg, median (Q1, Q3)	70 (70, 80)	70 (70, 80)	70 (65, 80)	0.516
Body-temperature at discharge, Celsius, median (Q1, Q3)	36.0 (36.0, 36.2)	36.0 (36.0, 36.2)	36.0 (36.0, 36.5)	0.09
Hemoglobin at discharge, g/dL, median (Q1, Q3)	12.1 (10.7, 13.2)	12.1 (10.8, 13.2)	11.6 (10.0, 12.7)	0.103
C-reactive protein at discharge, mg/dL, median (Q1, Q3)	0.7 (0.2, 2.0)	0.6 (0.2, 1.9)	0.9 (0.2, 3.2)	0.787
White blood cell count at discharge, 10^3^/mm^3^, median (Q1, Q3)	6.59 (5.24, 8.10)	6.46 (5.24, 8.05)	7.43 (6.83, 8.46)	0.066

Q1, first quartile; Q3, third quartile; CI, confidence interval; bpm, beats per minute; COVID, COronaVIrus-related Disease; * Chi-square test for categorical variables, Mann-Whitney test for continuous variables.

**Table 2 jcm-10-05590-t002:** Distribution of cardio-circulatory and biochemical variables at admission, at discharge and their variation.

	At Admission	At Discharge	Mean Variation * (95% CI)
Heart rate, bpm, median (Q1, Q3)	88 (78, 101)	79 (70, 88)	−10 (−12, −9)
Mean (SD)	90 (18)	80 (2)	
Oxygen saturation, %, median (Q1, Q3)	96 (93, 98)	97 (96, 98)	+1.9 (+1.6, +2.2)
Mean (SD)	95 (4)	97 (2)	
Systolic blood pressure, mmHg, median (Q1, Q3)	130 (120, 149]	120 (110, 130]	−12 (−14, −10)
Mean (SD)	133 (23)	122 (15)	
Diastolic blood pressure, mmHg, median (Q1, Q3)	75 (70, 80)	70 (70, 80)	−2 (−3, −1)
Mean (SD)	75 (12)	73 (9)	
Body-temperature, Celsius, median (Q1, Q3)	36.8 (36.1, 37.7)	36.0 (36.0, 36.2)	−0.9 (−1.0, −0.8)
Mean (SD)	37.0 (1.0)	36.1 (0.4)	
Hemoglobin, g/dL, median (Q1, Q3)	13.5 (12.1, 14.5)	12.1 (10.7, 13.2)	−1.2 (−1.3, −1.1)
Mean (SD)	13.2 (1.9)	12.0 (1.8)	
C-reactive protein, mg/dL, median (Q1, Q3)	4.6 (1.4, 9.67)	0.7 (0.2, 2.0)	−5.1 (−5.6, −4.5)
Mean (SD)	6.7 (6.9)	1.8 (2.8)	
White blood cell count, 10^3^/mm^3^, median (Q1, Q3)	6.64 (4.85, 9.25)	6.59 (5.24, 8.10)	−0.64 (−0.95, −0.34)
Mean (SD)	7.66 (4.73)	7.06 (3.94)	

Q1, first quartile; Q3, third quartile; CI, confidence interval; bpm, beats per minute; * calculated as value at discharge minus value at admission.

**Table 3 jcm-10-05590-t003:** Predictors of heart rate at discharge in hospitalized patients with SARS-CoV-2 infection.

	Models Adjusted Only by Heart Rate at Admission		Fully Adjusted Model *	
	Coefficient (95% CI)	*p*	Coefficient (95% CI)	*p*
Heart rate at admission, per bpm increase	0.17 (0.13; 0.23)	<0.001	0.17 (0.11; 0.22)	<0.001
Age, per year increase	−0.10 (−0.16; −0.04)	<0.001	−0.03 (−0.12; 0.06)	0.511
Gender, female as reference	0.15 (−1.78; 2.08)	0.878	−0.08 (−2.11; 1.94)	0.935
Charlson Comorbidity Index, per unit increase	−0.77 (−1.13; −0.41)	<0.001	−0.36 (−0.91; 0.19)	0.195
Oxygen saturation, %, per unit increase	−0.23 (−0.48; −0.02)	0.073	−0.09 (−0.36; 0.18)	0.506
Systolic blood pressure, per 1 mmHg increase	−0.041 (−0.081; −0.001)	0.046	−0.010 (−0.051; 0.032)	0.640
Body-temperature Celsius, per unit increase	0.19 (−0.80; 1.18)	0.703	−0.47 (−1.47; 0.52)	0.348
Hemoglobin, per g/dL increase	−0.52 (−1.02; −0.03)	0.039	−0.64 (−1.19; −0.09)	0.023
Severe disease, absence as reference	9.64 (6.94; 12.35)	<0.001	8.42 (5.39; 11.45)	<0.001
COVID-19 pulmonary disease, absence as reference	1.58 (−1.05; 4.22)	0.239	1.36 (−1.48; 4.20)	0.348
Drugs acting on heart rate, absence as reference	−2.67 (−4.73; −0.62)	0.011	−1.84 (−4.01; 0.33)	0.097

COVID, COronaVIrus-related Disease; * Intercept (std. error) = 101.492 (24.632), adjusted R2 = 13.3%.

## Data Availability

Available on request.

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
