# Peer review of "Heart Rate in Patients with SARS-CoV-2 Infection: Prevalence of High Values at Discharge and Relationship with Disease Severity"

_jcm, 2021, doi:10.3390/jcm10235590_

Round 1
Reviewer 1 Report
Thank you for asking me to review this. I have found this article to be well written and the study well conducted. It is a simple observational study from which conclusions can be drawn but not much correlation. However, I still think that this article is worthy of publication in its current form. The methods were well thought of and the results are well presented
I would like the authors to
- correct typos. for example line 286 i do not think the word they are looking for is interfere? and scattered through the manuscript are others like data 'is' etc
- add a section in bullet points for future points for research
- add a section as to why they thought the study was important at the time they started it- at that time, most of us were just dealing with a new disease and struggling, and thus to have the foresight to do this was very good
Many thanks again
Author Response
We have to thank the reviewer for the time and energy that invested in our manuscript. We believe that the comments and suggestions provided in the revision, gave us the opportunity to improve the manuscript considerably and better present the results of our study. Please find below the answers to all of the comments point by point. The respective changes have been highlighted with the track changes in the revised version. We hope that the revised manuscript meets also the expectations of the reviewer.
Thank you for asking me to review this. I have found this article to be well written and the study well conducted. It is a simple observational study from which conclusions can be drawn but not much correlation. However, I still think that this article is worthy of publication in its current form. The methods were well thought of and the results are well presented. I would like the authors to
- correct typos. for example line 286 i do not think the word they are looking for is interfere? and scattered through the manuscript are others like data 'is' etc
Following this suggestion paper was carefully assesses for typos and English grammar.
- add a section in bullet points for future points for research
Suggested section have been added after formal conclusions.
- add a section as to why they thought the study was important at the time they started it- at that time, most of us were just dealing with a new disease and struggling, and thus to have the foresight to do this was very good
We must thank again the reviewer that with this comment allow us to better explain the aim of our study and how it is developed. This study arises from our clinical experience on COVID-19 patients in which we observed a high prevalence of sinus tachycardia and elevated discharge HR that could also persist over time. This led us to design this analysis on patients discharged from our hospital and to try to understand why some patients experience this problem and some others not. This is now better stated in the final paragraph of the introduction (line 66).
Reviewer 2 Report
The manuscript is well written, even if more attention should be paid to English grammar and structure. The study is well designed, conducted, and presented; therefore, the Authors should be commended for this work. To increase the manuscript's soundness, clarity, and readability, I suggest changing all the P-Values presented as 1 to 0.99.
IQR generally refers to a single value (the 25th percentile subtracted to the 75th one) and not to a range, as wrongly believed from the most. If you intended to display the 25th-75th percentile of a median distribution, suggest modifying IQR to Q1, Q3, or specifying in the "Statistical analysis" session that IQR refers to this interval.
I finally recommend a light English grammar revision and a second check for typos.
Author Response
We have to thank the reviewer for the time and energy that invested in our manuscript. We believe that the comments and suggestions provided in the revision, gave us the opportunity to improve the manuscript considerably and better present the results of our study. Please find below the answers to all of the comments point by point. The respective changes have been highlighted with the track changes in the revised version. We hope that the revised manuscript meets also the expectations of the reviewer.
The manuscript is well written, even if more attention should be paid to English grammar and structure. The study is well designed, conducted, and presented; therefore, the Authors should be commended for this work. To increase the manuscript's soundness, clarity, and readability, I suggest changing all the P-Values presented as 1 to 0.99.
All the p-values actually presented as 1 have been changed to 0.99.
IQR generally refers to a single value (the 25th percentile subtracted to the 75th one) and not to a range, as wrongly believed from the most. If you intended to display the 25th-75th percentile of a median distribution, suggest modifying IQR to Q1, Q3, or specifying in the "Statistical analysis" session that IQR refers to this interval.
IQR has been modified to Q1 and Q3 as suggested.
I finally recommend a light English grammar revision and a second check for typos.
Following this suggestion paper was carefully assesses for typos and English grammar.
This manuscript is a resubmission of an earlier submission. The following is a list of the peer review reports and author responses from that submission.
Round 1
Reviewer 1 Report
The authors showed HR in patients with COVID-19, relationship between HR at discharge and disease severity. The reviewer would like to raise some questions.
Major comments
- It is unclear what the HR at hospital discharge means accurately. I think that the statistically significant difference and the clinical usefulness are different.
- The authors stated that the HR at discharge correlates with the severity of COVID-19. However, it is desirable to consider what kind of prognostic factors the HR can be.
Minor comments
- The author should describe a comparison of SpO2, Body temperature, Blood pressure, Hb, CRP, WBC at discharge in table 3.
- Were there any patients with DVT or PE?
- The authors should also state whether they have used steroids such as dexamethasone.
- Why did so many patients take medications that control HR?
- Are you measuring HR or Pulse rate?
Reviewer 2 Report
The authors report the prevalence (5.5%) of high HR values in COVID subjects at hospitaldischarge.
The topic is interesting, because it remains unclear whether Heart Rate is a marker of response to sepsis or a parameter related to a long-term autonomic dysfunction.
Main point:
The data presented in Table 1 do not match those reported in Table 3; that is, the total numer of subjects is 701 in tab 1 and 697 in tab 3. Similarly other values do not match.
A major limitation is the absence of follow-up of some patients that might allow defining whether HR can persist after discharge and represent a clinical parameter after severe disease. The correlation oh Heart Rate and SARS CoV-2 infection has not been well characterized.
Table 1 could be cut and features of all patients could be included in Table 3.
Minor points:
There are some typos in the text and english should be improved, particularly in the ‘Discussion’
Round 2
Reviewer 1 Report
It is unclear what the HR at discharge means accurately. The autoimmune dysfunction is an interesting idea, but I think there is a lack of papers that strongly suggest this point.
Due to the small number of patients, no mention of differential diagnosis such as hyperthyroidism, and no long-term follow up data after discharge, it is difficult to show the importance.
Reviewer 2 Report
There are still grammatical errors i.e. if this have... line 302; analysis were corrected.... line 286
Please, check the text